# Cost-effectiveness of virtual emergency care models: A protocol for a systematic review

Ravi Shankar[1]*, Linda Wang[2], Ho Soon Hoe[2], Liew Mei Fong[2,3],
Satya Pavan Kumar Gollamudi[4,5], Serene Wong[2,3,4]

1 Medical Affairs—Research Innovation & Enterprise, Alexandra Hospital, Singapore, 2 Fast and Chronic Programmes, Alexandra Hospital, Singapore, 3 Division of Respiratory and Critical Care Medicine, Department of Medicine, National University Hospital, Singapore, 4 Yong Loo Lin School of Medicine, National University of Singapore, Singapore, 5 Division of Advanced Internal Medicine, Department of Medicine, National University Hospital, Singapore

* Ravi_SHANKAR@nuhs.edu.sg

## Abstract

### Background

The COVID-19 pandemic has accelerated the adoption of virtual care models in emergency medicine. While virtual emergency care has the potential to expand access, improve efficiency, and reduce costs, rigorous evaluation of its cost-effectiveness compared with traditional in-person emergency care is needed.

### Objective

This systematic review aims to comprehensively search the literature, critically appraise the evidence, and synthesize findings on the cost-effectiveness of virtual emergency care models compared to in-person emergency care.

### Methods

We will search PubMed, Web of Science, Embase, CINAHL, MEDLINE, The Cochrane Library, PsycINFO, and Scopus from 2010 to February 2025 for economic evaluations that report both costs and effects comparing virtual and in-person emergency care models. Studies that compare multiple virtual interventions without an in-person care comparator will be excluded. Two reviewers will independently screen studies, extract data, and assess methodological quality and risk of bias using established quality assessment tools. Covidence software will be used to manage the screening and data extraction process. A narrative synthesis and quantitative meta-analysis of incremental cost-effectiveness ratios (ICERs) will be conducted if appropriate.

**Data availability statement:** No datasets were generated or analysed during the current study. All relevant data from this study will be made available upon study completion.

**Funding:** The author(s) received no specific funding for this work.

**Competing interests:** The authors have declared that no competing interests exist.

## Discussion

This review will provide a comprehensive evidence synthesis on the cost-effectiveness of virtual emergency care to guide clinical implementation, health policy, and future research. Findings will be highly relevant as virtual care becomes increasingly integrated into emergency care delivery in the aftermath of the COVID-19 pandemic.

## PROSPERO registration

CRD42025648218

## Introduction

Emergency departments (EDs) worldwide face persistent challenges of overcrowding, prolonged wait times, and escalating healthcare costs [1–3]. ED crowding is associated with treatment delays, medical errors, patient dissatisfaction, and increased morbidity and mortality [4,5]. Concurrently, the rising costs of emergency care strain patients, insurers, and health systems [6]. In 2017, US EDs accounted for 4.4% of total US health expenditures, or $76.3 billion [7,8].

Virtual emergency care, encompassing telemedicine, remote monitoring, and digital triage, has been increasingly proposed as a strategy to mitigate these challenges by expanding access, improving efficiency, and reducing costs [9–11]. Virtual ED consultations enable remote assessment and management of conditions that may not require in-person care, through synchronous (e.g., video) or asynchronous (e.g., text) digital interactions between patients and emergency providers [12,13]. Remote provider-to-provider teleconsultations, such as telestroke or teletrauma, connect ED providers with specialists to guide time-critical decisions [14]. Digital triage tools, such as web-based symptom checkers or chatbots, provide automated risk assessment, self-care guidance, or ED referral [15]. Remote physiological monitoring devices enable continuous tracking and alerts for specific high-risk conditions such as sepsis or acute heart failure [16,17].

The COVID-19 pandemic catalyzed a rapid uptake of virtual emergency care models around the world [18,19]. For example, in the US, ED telemedicine visits increased 16-fold between January and July 2020 compared with the same time period in 2019 [20]. The pandemic experience highlighted the potential for virtual modalities to facilitate triage, conserve protective equipment, reduce exposure risks, and preserve ED capacity [21]. Yet it also exposed barriers related to technology access, digital literacy, health equity, diagnostic uncertainty, and information security [22,23].

As the pandemic subsides, health systems and emergency care providers must decide whether and how to sustain virtual models as part of the standard practice [24]. A key consideration is cost-effectiveness – whether virtual emergency care provides sufficient value compared to traditional in-person care to justify ongoing investments. Some economic models suggest that virtual ED triage and consultations can reduce costs by diverting low-acuity visits, or by substituting for costlier in-person

care [3,25,26]. However, there are also concerns that virtual visits may duplicate rather than replace in-person visits, increasing overall costs and utilization [27].

Prior systematic reviews on virtual emergency care have focused on clinical effectiveness or implementation outcomes [28–30], with limited evaluations of economic impact. Existing economic reviews have focused on specific conditions like acute stroke [31], or specific technologies like wearables [32], with less attention to system-wide emergency care costs. To inform post-pandemic emergency care redesign, a comprehensive synthesis of economic evidence on virtual emergency care is urgently needed.

A preliminary scoping search conducted in PubMed identified approximately 150 potentially relevant studies, suggesting sufficient literature exists to warrant this comprehensive systematic review. The scoping search revealed heterogeneity in virtual care modalities, outcome measures, and economic evaluation methods, supporting our planned narrative synthesis approach.

## Objective

This systematic review aims to evaluate the cost-effectiveness of virtual emergency care models compared to traditional in-person emergency care. While our primary focus is on economic outcomes, we will extract and report implementation barriers and facilitators as secondary findings when reported in included economic evaluations.

## Methods

This systematic review will be conducted and reported according to the Preferred Reporting Items for Systematic review and Meta-Analysis (PRISMA) guidelines [33] and the Consolidated Health Economic Evaluation Reporting Standards (CHEERS) [34].

### Eligibility criteria

We will select studies according to the following PICOS (Population, Intervention, Comparator, Outcomes, Study design) criteria.

**Population.** Patients of any age with emergency conditions that are the focus of a virtual care intervention. Emergency conditions are defined as acute illnesses or injuries that require immediate medical attention to prevent death or disability, as assessed by triage personnel, emergency providers, or patients themselves. Both undifferentiated emergency presentations (e.g., chest pain) and specific diagnoses (e.g., acute stroke) will be included. We will exclude studies on non-emergent conditions, elective procedures, or hospital inpatients without preceding ED care.

**Intervention.** Any acute care delivery model that uses virtual modalities to manage emergency conditions as an alternative or adjunct to in-person emergency care. Virtual modalities include:

- Synchronous or asynchronous telemedicine consultations between patient and emergency provider

- Remote provider-to-provider telemedicine consults between ED and specialists

- Web or mobile-based triage tools for symptom assessment, self-care guidance or disposition

- Wearable devices or remote monitoring systems for high-risk emergency conditions

Hybrid models that combine initial virtual assessment with selective in-person care (e.g., mobile stroke units) will be included.

**Comparator.** Usual in-person emergency care, delivered through face-to-face patient-provider interactions in a hospital-based emergency department. This may involve an initial in-person triage assessment by a nurse, and subsequent in-person evaluation and management by an emergency physician or advanced practice provider.

**Outcomes.** The primary outcome is the incremental cost-effectiveness ratio (ICER), calculated as the difference in mean costs between virtual and in-person models divided by the difference in mean effectiveness. Costs should be reported from the health system or societal perspective, inflated to a common currency and year. Effectiveness should be measured using validated clinical (e.g., mortality), patient-centered (e.g., quality of life), or process (e.g., ED throughput) outcomes. Secondary outcomes include incremental costs and effects (even when not reported as ICERs), net monetary benefit, and sensitivity analyses.

**Study designs.** We will include full economic evaluations that compare both costs and effects of virtual and in-person emergency care models. Eligible designs include cost-effectiveness analysis (using clinical or process outcomes), cost-utility analysis (using quality-adjusted life years or QALYs), cost-benefit analysis (monetizing health outcomes), and cost-minimization analysis (assuming equal effectiveness). Both model-based and trial-based evaluations will be included.

**Exclusion criteria.** We will exclude studies that:

- Do not involve an emergency condition or emergency care setting

- Compare multiple virtual interventions without an in-person care comparator

- Do not report both costs and effectiveness outcomes

- Are not original research (e.g., reviews, editorials, commentaries)

- Are published before 2010, as virtual care technologies have advanced significantly in the last decade

**Geographic and language scope.** Studies from all countries will be included to capture the global evidence base on virtual emergency care cost-effectiveness. No language restrictions will be applied, ensuring that relevant studies published in any language are considered for inclusion. Efforts will be made to translate non-English studies where necessary to facilitate comprehensive analysis.

## Information sources and search strategy

We will conduct a comprehensive search across multiple electronic bibliographic databases from 2010 to February 2025. The databases included in our search strategy are PubMed, Web of Science, Embase, CINAHL, MEDLINE, The Cochrane Library, PsycINFO, and Scopus. This extensive approach ensures a thorough and systematic review of the relevant literature, capturing a wide range of studies across various disciplines.

A search strategy will be developed for each database using a combination of keywords and controlled vocabulary terms (e.g., MeSH) related to (1) emergency care, (2) virtual care models, and (3) economic evaluations. An example of a search string is:

((virtual OR telehealth OR telemedicine OR digital OR "remote consult*" OR mHealth OR eHealth OR "mobile health" OR "web-based" OR wearable* OR "remote monitoring") AND (emergency OR emergencies OR ED OR ER OR "emergency department") AND ("cost effectiv*" OR "cost utility" OR "cost benefit" OR "cost minimization" OR "economic evaluation"))

Additional search filters will be applied for publication date (2010-present). We will also hand-search reference lists of included studies and relevant reviews for additional eligible studies. Grey literature sources such as clinical trial registries, conference proceedings, and preprint servers will be searched to identify unpublished studies. The full search strategy for each database will be developed in consultation with a medical librarian.

## Study selection

Two reviewers will independently screen titles and abstracts of search results against the pre-specified PICOS criteria using Covidence systematic review software [35]. Records will be classified as "yes", "no" or "maybe" for inclusion. The full text of potentially eligible studies classified as "yes" or "maybe" by either reviewer will be retrieved for further assessment.

The two reviewers will then independently review the full text of these studies using a standardized screening form with the detailed PICOS criteria. Reasons for exclusion will be recorded. Disagreements will be resolved through discussion or arbitration by a third reviewer.

The study selection process will be documented in a PRISMA flow diagram indicating the number of studies identified, included and excluded, with reasons for exclusion. The full list of studies excluded after full-text review will also be provided.

### Data extraction

A standardized data extraction form will be developed based on the CHEERS reporting standards and piloted on a sample of five included studies. Two reviewers will independently extract data from all included studies using Covidence, with any discrepancies resolved through consensus or by involving a third reviewer. The extracted data will cover several key areas, including bibliographic details, participant characteristics, interventions, comparators, outcomes, and study-specific details.

Bibliographic details will include the first author, publication year, journal, and funding sources. Participant characteristics will capture inclusion and exclusion criteria, sample size, age (mean, standard deviation, range), sex distribution (% female), emergency conditions, triage acuity levels, and comorbidities. Intervention data will focus on the virtual care model used (e.g., ED-based telemedicine, teletriage, remote monitoring), the technological medium (e.g., video, telephone, web, mobile app), provider type (e.g., physician, nurse, advanced practice provider), frequency and duration of the intervention, and its relationship to in-person care (e.g., standalone, triage, step-up/step-down models).

Comparator details will include a description of usual in-person emergency care, specifying the providers, processes, and resources involved. Outcomes will be assessed through measures such as the incremental cost-effectiveness ratio (ICER) and its components, the type and source of effectiveness data (e.g., QALYs from trials, ED length of stay from administrative data), and the type and source of cost data (e.g., payer, hospital, patient costs). Other economic evaluation factors will be documented, including the analysis perspective (e.g., healthcare system, societal), time horizon, discount rate, sensitivity analyses (e.g., probabilistic, deterministic), and subgroup analyses.

Study details will cover the country and setting, analytic approach (e.g., within-trial, model-based), model structure and assumptions, utility scores and sources, cost sources and currency, and the approach to handling missing data (Details in Appendix 1 S1 File).

### Risk of bias assessment

The methodological quality and risk of bias of included economic evaluations will be independently assessed by two reviewers using the Drummond 10-item quality assessment tool and the Consensus Health Economic Criteria (CHEC) 19-item quality assessment tool [36–38]. The Drummond tool evaluates key aspects such as clarity of research questions, description of alternatives, effectiveness evidence, cost and consequence identification, valuation, adjustment for time differences, incremental analysis, uncertainty consideration, and user relevance. The CHEC tool assesses study population clarity, research question formulation, appropriateness of study design and time horizon, perspective selection, cost and outcome identification, measurement and valuation, incremental analysis, discounting, sensitivity analysis, result generalizability, conflict of interest disclosure, and ethical considerations (Details in S1 File).

The quality assessment results will be reported in tables with color coding for risk of bias: green (low), yellow (some concerns), or red (high). Results will also be summarized narratively and inform the overall quality of the evidence (e.g., using GRADE economic criteria) [39].

### Synthesis and analysis

A narrative synthesis, following the Synthesis Without Meta-analysis (SWiM) guidelines [40], will first summarize key characteristics and findings of included economic evaluations in text and tabular form. Tables will report details on populations, interventions, comparators, analytic approach, cost and outcome measures, and ICERs. We will compare

cost-effectiveness results across studies and explore potential reasons for variability, such as differences in health systems, methodological choices, or evaluative scope.

In cases of significant methodological heterogeneity across included studies, we will employ a structured approach to narrative synthesis. Studies will be categorized by virtual care modality (e.g., synchronous video, asynchronous messaging, remote monitoring), clinical context (e.g., undifferentiated ED presentations, specific conditions like stroke), and economic evaluation type. Within each category, we will identify patterns in cost-effectiveness findings and contextualize results based on healthcare setting, patient population, and implementation factors. When outcome measures differ substantially, we will standardize results to common metrics where possible (e.g., converting to cost per quality-adjusted life year) or focus on reporting relative rather than absolute economic outcomes [41,42].

If there are two or more studies with sufficiently homogeneous populations, interventions and methods, we will pool ICERs using fixed or random-effects meta-analysis [43]. Heterogeneity will be assessed statistically (e.g., $I^2$) and through subgroup analyses based on key study-level characteristics (e.g., virtual care model, patient acuity, country). Sensitivity analyses will examine the impact of study quality. Publication bias will be assessed visually through funnel plots and statistically using Egger's test [44].

If meta-analysis is not appropriate, we will focus on describing economic findings narratively and identifying key drivers of cost-effectiveness across studies. We will also summarize implementation issues, evidence gaps, and methodological challenges to guide future economic evaluation efforts in virtual emergency care.

All analyses will be conducted in R version 4.0.

### Confidence in cumulative evidence

We will apply the GRADE approach to rate the certainty of economic evidence for each outcome as high, moderate, low, or very low [45]. Ratings will be based on risk of bias, imprecision, inconsistency, indirectness, and publication bias assessments. The implications of these ratings for decision-making will be discussed.

GRADE ratings will directly inform how findings can guide health policy decisions. For outcomes with high-certainty evidence, we will provide specific implementation recommendations. For moderate-certainty evidence, we will suggest consideration with appropriate monitoring mechanisms. For low or very-low certainty evidence, we will emphasize the need for caution and additional research before widespread implementation. GRADE criteria will be applied to all included studies regardless of methodological quality threshold, with sensitivity analyses conducted to assess the impact of excluding lower-quality studies on overall conclusions.

### Stakeholder engagement

Representatives from key stakeholder groups (emergency physicians, health economists, hospital administrators, payers, and patient advocates) will be engaged early in the review process to inform data extraction priorities, outcome interpretation, and knowledge translation planning. This engagement will occur through structured consultations during protocol refinement and preliminary findings review.

### Living systematic review approach

This systematic review will be maintained as a living review, with planned updates conducted annually or when substantial new evidence becomes available. Updated searches will use the same methodology, with screening and data extraction conducted by the same research team to ensure consistency. Stakeholders will be notified of significant updates through our knowledge translation networks [46].

### Discussion

This protocol outlines a systematic and comprehensive approach to evaluating the cost-effectiveness of virtual emergency care models compared to traditional in-person care. The COVID-19 pandemic has accelerated the adoption of virtual

care in emergency medicine, but the economic value of these models remains uncertain. This systematic review aims to address this evidence gap by synthesizing the highest quality economic evaluations available to inform clinical implementation, health policy, and future research directions.

The protocol ensures methodological rigor which is a key strength of the systematic review. By adhering to best practice guidelines for systematic review (PRISMA), health economic evaluations (CHEERS), and synthesis without meta-analysis (SWiM), the protocol ensures transparency, reproducibility, and sound methodology. The inclusion and exclusion criteria are clearly specified using the PICOS framework, covering a broad range of virtual care modalities, emergency conditions, and economic outcomes. The search strategy is comprehensive, leveraging both academic databases and grey literature sources, and will be iteratively developed in consultation with a medical librarian to optimize sensitivity and specificity.

The planned use of independent dual review and standardized data extraction forms adapted from the CHEERS reporting standards aim to minimize the risk of bias and errors in study selection and data collection. Assessment of methodological quality and risk of bias using the Drummond and CHEC checklists will provide a transparent evaluation of the internal and external validity of included studies. Results will inform the certainty of evidence ratings using the GRADE approach. If appropriate, quantitative synthesis using meta-analysis will pool ICERs to provide summary estimates of cost-effectiveness, with exploration of heterogeneity through subgroup and sensitivity analyses, as demonstrated by Noparatayaporn et al. [47], which evaluated the cost-effectiveness of bariatric surgery across different patient subgroups and time horizons through a meta-analysis of incremental net monetary benefit. If quantitative synthesis is not possible, narrative synthesis following SWiM guidelines will summarize key findings, explore heterogeneity across studies, and identify implications for policy and research.

Anticipated limitations of the review largely reflect the current state of evidence in this rapidly evolving field. The COVID-19 pandemic catalyzed numerous virtual emergency care initiatives and research studies, but many are likely still ongoing or unpublished. The lag in dissemination of findings may result in a skewed evidence base that over-represents earlier, less mature virtual care models and fails to capture longer-term economic outcomes. The inclusion of pre-prints and grey literature aims to mitigate publication bias, but the planned assessment of publication bias will be important to contextualize the findings.

Another challenge relates to the significant heterogeneity expected across studies in terms of populations, interventions, comparators, methods, and contexts. Variability in virtual care modality, technological platform, staffing models, clinical protocols, and payment structures may preclude fair comparisons of cost-effectiveness across studies. Inconsistency in costing methods, cost and outcome measures, time horizons, and analytic perspectives may further limit quantitative pooling of ICERs. However, the use of established economic evaluation reporting standards (CHEERS) and quality assessment tools (Drummond, CHEC) to critically appraise studies and a narrative synthesis approach (SWiM) to explore heterogeneity can still yield valuable insights to guide decision-making.

Next steps following this protocol involve executing the systematic review according to the pre-specified methods. This will require close collaboration among a multidisciplinary team of emergency clinicians, health economists, systematic review methodologists, and informatics experts. Adaptations to the protocol may be necessary based on the volume and characteristics of the evidence base, with any deviations transparently reported.

Findings will be rapidly disseminated through academic publications, conference presentations, and briefs targeted to key emergency care stakeholders, including healthcare providers, payers, policymakers, patient advocates, and technology vendors. Plain-language summaries, infographics, and evidence maps can enhance the accessibility and impact of results for diverse audiences. Insights will be contextualized to different health systems and policy environments to facilitate local adaptation and implementation.

To maximize real-world application beyond academic audiences, findings will be translated into targeted knowledge products for key stakeholders: (1) policy briefs for health ministries and regulatory bodies highlighting cost-effectiveness

thresholds and policy implications; (2) implementation toolkits for hospital administrators with economic modeling templates and budget impact analyses; (3) strategic reports for telehealth providers identifying sustainable reimbursement models; (4) value-based payment frameworks for insurance companies and payers; and (5) decision aids for patients and consumer advocates explaining economic trade-offs and access considerations. We will engage representatives from each stakeholder group to ensure these knowledge translation products address practical decision-making needs.

In parallel, a key research priority will be to update this systematic review on an ongoing basis as new economic evaluations of virtual emergency care models are published. Living systematic reviews can help keep pace with the rapidly evolving evidence base and inform real-time decision-making. Establishing a shared repository for standardized reporting of economic outcomes for virtual emergency care can further facilitate evidence synthesis and cross-study comparisons.

Ultimately, this systematic review aims to provide actionable economic evidence to guide the implementation of virtual emergency care models that enhance patient access, improve population health outcomes, and lower costs. By identifying cost-effective models and implementation best practices, the review can accelerate evidence-based integration of virtual care into routine emergency care delivery. At a health system level, this can support progress towards the quadruple aim of better care experiences, better population health, lower costs, and improved provider well-being [48,49]. Demonstrating the economic value of virtual emergency care can also help secure sustainable funding, incentives, and infrastructure to support ongoing telehealth innovation and evaluation in emergency medicine.

This systematic review protocol represents a rigorous and comprehensive approach to synthesizing economic evidence on virtual emergency care, a rapidly evolving area with significant policy and practice implications in the post-pandemic era. Adherence to best practice guidelines for systematic reviews and health economic evaluations strengthens the methodological quality, transparency, and reproducibility of the planned review. The results will equip health system leaders, payers, and policymakers with critical economic insights to guide evidence-based integration of cost-effective virtual emergency care models. Findings will also highlight priority areas for future economic evaluations to strengthen the evidence base on virtual emergency care. As both an emergency physician and health services researcher dedicated to telehealth innovation, I believe this systematic review is a crucial step towards realizing the potential of virtual care in emergency medicine to enhance the quadruple aim and advance health system value.

## Supporting information

**S1 Checklist. PRISMA-P Checklist.**
(DOC)

**S1 File. Supplementary file.**
(DOCX)

## Author contributions

**Conceptualization:** Ravi Shankar.

**Data curation:** Ravi Shankar, Ho Soon Hoe.

**Investigation:** Ravi Shankar.

**Methodology:** Ravi Shankar, Linda Wang.

**Project administration:** Ravi Shankar, Linda Wang.

**Resources:** Ravi Shankar, Serene Wong.

**Software:** Ravi Shankar, Serene Wong.

**Supervision:** Ravi Shankar, Linda Wang, Liew Mei Fong, Satya Pavan Kumar Gollamudi, Serene Wong.

**Validation:** Ravi Shankar, Linda Wang, Ho Soon Hoe.

**Visualization:** Linda Wang, Ho Soon Hoe.

**Writing – original draft:** Ravi Shankar.

**Writing – review & editing:** Ravi Shankar, Linda Wang, Ho Soon Hoe, Liew Mei Fong, Satya Pavan Kumar Gollamudi, Serene Wong.

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
