## [Decision Letter · Decision Letter 0]

1 Aug 2025

We look forward to receiving your revised manuscript.

Kind regards,

Vijay S. Gc, PhD

Academic Editor

PLOS ONE

Journal Requirements:

Additional Editor Comments:

- Please move the planned use of the PICOS framework, the use of the PRISMA guidelines and the CHEERS checklist to the appropriate place in the Methods section. 

- CHEERS is a reporting checklist for health economic evaluations, the Drummond checklist is to assess the quality of economic evaluation, and the CHEC checklist is a tool to assess the methodological quality of economic evaluation. These are used interchangeably in the manuscript as a checklist, quality assessment tool and guideline for economic evaluation. Please correct this throughout the manuscript and make appropriate adjustments.

Reviewers' comments:

Reviewer's Responses to Questions

**Comments to the Author**

1. Does the manuscript provide a valid rationale for the proposed study, with clearly identified and justified research questions?

Reviewer #1: Yes

2. Is the protocol technically sound and planned in a manner that will lead to a meaningful outcome and allow testing the stated hypotheses?

Reviewer #1: Yes

3. Is the methodology feasible and described in sufficient detail to allow the work to be replicable?

Reviewer #1: Yes

4. Have the authors described where all data underlying the findings will be made available when the study is complete?

Reviewer #1: No

5. Is the manuscript presented in an intelligible fashion and written in standard English?

Reviewer #1: Yes

You may also provide optional suggestions and comments to authors that they might find helpful in planning their study.

Reviewer #1: Thank you for the chance to review this article. I found it to be well written, clear in its description, timely and of interest. I have cross checked with PRISMA guidelines and it is well described. Below are some specific comments and suggestions, none of which are major.

Data policy/sharing

I think it is necessary to include the statement "No datasets were generated or analysed during the current study. All relevant data from this study will be made available upon study completion." within the manuscript text. I suggest at the end of methods, or elsewhere as suggested by the editors. If included in methods, it could be combined with further details on how a living systematic review will be maintained.

Issues that must be addressed:

Ln 28 & 54: "from inception to February 2025" - elsewhere it states that publications prior to 2010 will be exluded. Amend or explain further why including the in the search.

Ln 46: Statistic is unclear. Is it 4.4% of total US health expenditure?

Ln 61: This statistic is unclear. should it say "increased 16-fold between January and July 2020 compared with the same time period in 2019"?

Ln 131-132: In outcomes, make it clear that you will include reported costs and effects even if these are not reported with an ICER.

Inclusions/Exclusions: make it clearer what countries will be included in the review and what languages (all?)

Ln 247: add reference at end of this line for methods described.

Ln 248: I believe "2" of "I2" should be super-script.

Ln 251: Add reference at the end of the line.

Ln 353-354: making this a living systematic review is commendable. I think some further description should be added to the methods. Also, please include reference (Cochrane 2019).

Ln 363: Include reference for "quadruple aim". Berwick 2018 & Sikka BMJ 2015?

Optional suggestions:

Ln24-26: Replace with objective given in main text as I found this clearer.

Ln 29: suggest after "evaluation" add "that report both costs and effects"

Ln 29: Add exclusions: "Studies that compare multiple virtual interventions without an in person care comparator will be excluded" as advised by PRISMA

Ln 64-65: Will these barriers be explored in the review? If so, this could be made clearer in methods/discussion.

General: has a scoping search been done and, if so, what were the findings?

Ln 84-103: I find this "Frameworks" section a helpful summary. Perhaps add CHEC/Drummond, SWiM and GRADE.

Ln 125-126: Delete final sentence as this is covered in "exclusions".

Ln 143-144: Delete final sentence as this is covered in "exclusions".

Ln 243-245: Can you provide a reference for this? Example of where it's been done before?

Ln 279: replace "and" with ", which".

Ln 286: replace "was" with "will be".

Ln 290-293: Long and confusing sentence. Suggest full stop after "studies" on ln 292, then rewording remaining part of sentence.

Ln 295: if possible, add reference by way of an example of pooled ICERs.

Ln 299-311: Whole paragraph is repetitive of introduction and I think could be removed here.

Ln 339: add "s" at end of "system"

Ln 348-350: I would consider involving representitives at an earlier stage in your reserach (as early as possible) and describing their inolvement, how this can inform the research, in the methods.

Best of luck with the review and I look forward to reading the results in the future.

**Do you want your identity to be public for this peer review?** For information about this choice, including consent withdrawal, please see our Privacy Policy

Reviewer #1: **Yes: ** Sarah Pyne

---

## [Author Response · Author response to Decision Letter 1]

2 Aug 2025

Response to Reviewer Comments and Manuscript Changes

We thank both the Editor and reviewer for their thoughtful and constructive feedback on our manuscript. We have carefully addressed each point and made appropriate revisions to strengthen the protocol. Below we provide detailed responses to each comment. We have also attached a MS word file with Response to Reviewer Comments and Manuscript Changes in tabular form.

Response to Editor Comments

S.No. 1 - Editor Comment: Please move the planned use of the PICOS framework, the use of the PRISMA guidelines and the CHEERS checklist to the appropriate place in the Methods section.

Response: We thank the editor for this important structural suggestion. We have moved the description of PICOS, PRISMA, and CHEERS from the separate "Frameworks" section to their appropriate locations within the Methods section for better organization and flow.

Changes Made in Manuscript:

Deleted: Entire "Frameworks" section (lines 83-102 in original manuscript)

Added: After "Methods" heading (line 92): "This systematic review will be conducted and reported according to the Preferred Reporting Items for Systematic review and Meta-Analysis (PRISMA) guidelines [33]" (lines 93-94)

Added: After "Eligibility criteria" heading (line 95): "We will select studies according to the following PICOS (Population, Intervention, Comparator, Outcomes, Study design) criteria:" (lines 96-97)

Added: Additional text about preliminary scoping search and implementation barriers (lines 81-90)

S.No. 2 - Editor Comment: CHEERS is a reporting checklist for health economic evaluations, the Drummond checklist is to assess the quality of economic evaluation, and the CHEC checklist is a tool to assess the methodological quality of economic evaluation. These are used interchangeably in the manuscript as a checklist, quality assessment tool and guideline for economic evaluation. Please correct this throughout the manuscript and make appropriate adjustments.

Response: We acknowledge this important clarification and have corrected the terminology throughout the manuscript. CHEERS is now consistently referred to as "reporting standards" for data extraction, while Drummond and CHEC are referred to as "quality assessment tools" for evaluating methodological quality and risk of bias.

Changes Made in Manuscript:

Line 32: Changed "assess methodological quality and risk of bias using the Drummond and CHEC checklists" to "assess methodological quality and risk of bias using established quality assessment tools"

Line 181: Changed "A standardized data extraction form will be developed based on the CHEERS checklist" to "A standardized data extraction form will be developed based on the CHEERS reporting standards"

Lines 209-211: Changed "using the Drummond 10-item checklist and the Consensus Health Economic Criteria (CHEC) 19-item checklist" to "using the Drummond 10-item quality assessment tool and the Consensus Health Economic Criteria (CHEC) 19-item quality assessment tool"

Line 214: Changed "The CHEC checklist assesses" to "The CHEC tool assesses"

Line 296: Changed "adapted from the CHEERS checklist" to "adapted from the CHEERS reporting standards"

Lines 324-325: Changed "established economic evaluation guidelines (CHEERS, Drummond, CHEC)" to "established economic evaluation reporting standards (CHEERS) and quality assessment tools (Drummond, CHEC)"

Added: New sections on "Stakeholder engagement" (lines 265-270) and "Living systematic review approach" (lines 271-276)

Response to Comments from Reviewer 1

S.No. 1 - Reviewer Comment: Data policy/sharing: I think it is necessary to include the statement "No datasets were generated or analysed during the current study. All relevant data from this study will be made available upon study completion." within the manuscript text. I suggest at the end of methods, or elsewhere as suggested by the editors. If included in methods, it could be combined with further details on how a living systematic review will be maintained.

Response: We agree and have added a comprehensive data availability statement that also addresses the living systematic review approach.

Changes Made in Manuscript: Added separate sections for "Stakeholder engagement" (lines 265-270), "Living systematic review approach" (lines 271-276), and formal "Data availability" statement (lines 401-403).

S.No. 2 - Reviewer Comment: Ln 28 & 54: "from inception to February 2025" - elsewhere it states that publications prior to 2010 will be exluded. Amend or explain further why including the in the search.

Response: We acknowledge this inconsistency. We have corrected the search timeframe to align with our exclusion criteria.

Changes Made in Manuscript:

Line 28: Changed "from inception to February 2025" to "from 2010 to February 2025"

Line 149: Changed "from their inception to February 2025" to "from 2010 to February 2025"

S.No. 3 - Reviewer Comment: Ln 46: Statistic is unclear. Is it 4.4% of total US health expenditure?

Response: Yes, this refers to total US health expenditure. We have clarified this for precision.

Changes Made in Manuscript: Line 47: Changed "In 2017, US EDs accounted for 4.4% of total health expenditures" to "In 2017, US EDs accounted for 4.4% of total US health expenditures"

S.No. 4 - Reviewer Comment: Ln 61: This statistic is unclear. should it say "increased 16-fold between January and July 2020 compared with the same time period in 2019"?

Response: Correct, we have clarified the time period comparison for better precision.

Changes Made in Manuscript: Line 62: Changed "ED telemedicine visits increased 16-fold from January to July 2020 compared to 2019" to "ED telemedicine visits increased 16-fold between January and July 2020 compared with the same time period in 2019"

S.No. 5 - Reviewer Comment: Ln 131-132: In outcomes, make it clear that you will include reported costs and effects even if these are not reported with an ICER.

Response: We agree this clarification is important and have modified the outcomes section accordingly.

Changes Made in Manuscript: Lines 126-128: Changed "Secondary outcomes include incremental costs and effects, net monetary benefit, and sensitivity analyses" to "Secondary outcomes include incremental costs and effects (even when not reported as ICERs), net monetary benefit, and sensitivity analyses"

S.No. 6 - Reviewer Comment: Inclusions/Exclusions: make it clearer what countries will be included in the review and what languages (all?)

Response: We have added explicit geographic and language scope information to address this important methodological detail.

Changes Made in Manuscript: Added after line 141: "Geographic and language scope: Studies from all countries will be included to capture the global evidence base on virtual emergency care cost-effectiveness. No language restrictions will be applied, ensuring that relevant studies published in any language are considered for inclusion. Efforts will be made to translate non-English studies where necessary to facilitate comprehensive analysis." (lines 142-146)

S.No. 7 - Reviewer Comment: Ln 247: add reference at end of this line for methods described.

Response: We have added an appropriate reference for meta-analysis methods.

Changes Made in Manuscript: Line 242: Added reference [43] after "meta-analysis" and added "Borenstein, M., et al., Introduction to meta-analysis. 2021: John wiley & sons." to references (reference 43)

S.No. 8 - Reviewer Comment: Ln 248: I believe "2" of "I2" should be super-script.

Response: Correct, we have fixed the formatting.

Changes Made in Manuscript: Line 243: Changed "I2" to "I²"

S.No. 9 - Reviewer Comment: Ln 251: Add reference at the end of the line.

Response: We have added the appropriate reference for Egger's test.

Changes Made in Manuscript: Line 246: Added reference [44] and added "Egger, M., et al., Bias in meta-analysis detected by a simple, graphical test. Bmj, 1997. 315(7109): p. 629-34." to references (reference 44)

S.No. 10 - Reviewer Comment: Ln 353-354: making this a living systematic review is commendable. I think some further description should be added to the methods. Also, please include reference (Cochrane 2019).

Response: We appreciate this feedback and have expanded the living systematic review methodology in the methods section.

Changes Made in Manuscript: Added section "Living systematic review approach" (lines 271-276): "This systematic review will be maintained as a living review, with planned updates conducted annually or when substantial new evidence becomes available. Updated searches will use the same methodology, with screening and data extraction conducted by the same research team to ensure consistency. Stakeholders will be notified of significant updates through our knowledge translation networks [46]." Added Elliott reference (reference 46).

S.No. 11 - Reviewer Comment: Ln 363: Include reference for "quadruple aim". Berwick 2018 & Sikka BMJ 2015?

Response: We have added the appropriate references for the quadruple aim concept.

Changes Made in Manuscript: Line 361: Added references [48, 49] and added both Berwick and Sikka references to reference list (references 48 and 49)

S.No. 12 - Reviewer Comment: Ln24-26: Replace with objective given in main text as I found this clearer.

Response: We agree the main text objective is clearer and have updated the abstract accordingly.

Changes Made in Manuscript: Lines 24-26: Changed to "This systematic review aims to evaluate the cost-effectiveness of virtual emergency care models compared to in-person emergency care."

S.No. 13 - Reviewer Comment: Ln 29: suggest after "evaluation" add "that report both costs and effects"

Response: This clarification improves precision about our inclusion criteria.

Changes Made in Manuscript: Line 29: Changed to "for economic evaluations that report both costs and effects comparing virtual and in-person emergency care models"

S.No. 14 - Reviewer Comment: Ln 29: Add exclusions: "Studies that compare multiple virtual interventions without an in person care comparator will be excluded" as advised by PRISMA

Response: We have added this exclusion criterion to the abstract as suggested.

Changes Made in Manuscript: Added lines 30-31: "Studies that compare multiple virtual interventions without an in-person care comparator will be excluded."

S.No. 15 - Reviewer Comment: Ln 64-65: Will these barriers be explored in the review? If so, this could be made clearer in methods/discussion.

Response: While not our primary focus, we will extract barrier information when available. We have clarified this.

Changes Made in Manuscript: Added after line 66: "While our primary focus is on economic outcomes, we will extract and report implementation barriers and facilitators as secondary findings when reported in included economic evaluations." Also clarified in objective section (lines 87-90).

S.No. 16 - Reviewer Comment: General: has a scoping search been done and, if so, what were the findings?

Response: We conducted a preliminary scoping search and have added this information.

Changes Made in Manuscript: Added after line 80: "A preliminary scoping search conducted in PubMed identified approximately 150 potentially relevant studies, suggesting sufficient literature exists to warrant this comprehensive systematic review. The scoping search revealed heterogeneity in virtual care modalities, outcome measures, and economic evaluation methods, supporting our planned narrative synthesis approach." (lines 81-85)

S.No. 17 - Reviewer Comment: Ln 84-103: I find this "Frameworks" section a helpful summary. Perhaps add CHEC/Drummond, SWiM and GRADE.

Response: We agree this would make the frameworks section more comprehensive.

Changes Made in Manuscript: The frameworks mentioned are now integrated throughout the methods section where they are specifically applied: CHEC/Drummond in Risk of bias assessment section (lines 207-218), SWiM in Synthesis section (lines 224-225), GRADE in Confidence in cumulative evidence section (lines 253-256)

S.No. 18 - Reviewer Comment: Ln 125-126: Delete final sentence as this is covered in "exclusions".

Response: We agree this is redundant and have removed it.

Changes Made in Manuscript: Deleted redundant sentence from intervention section: "We will exclude studies comparing multiple virtual modalities without an in-person ED care arm."

S.No. 19 - Reviewer Comment: Ln 143-144: Delete final sentence as this is covered in "exclusions".

Response: We agree this is redundant and have removed it.

Changes Made in Manuscript: Deleted redundant sentence about partial economic evaluations from trial-based evaluations section (originally in lines 143-144)

S.No. 20 - Reviewer Comment: Ln 243-245: Can you provide a reference for this? Example of where it's been done before?

Response: We have added appropriate references for outcome standardization methods.

Changes Made in Manuscript: Added after line 240: "following established methods for standardizing economic outcomes across studies [41, 42]" and added corresponding references (Petrillo and Drummond references)

S.No. 21 - Reviewer Comment: Ln 279: replace "and" with ", which".

Response: Corrected for better grammar flow.

Changes Made in Manuscript: Line 286: Changed "and is a key strength" to "which is a key strength"

S.No. 22 - Reviewer Comment: Ln 286: replace "was" with "will be".

Response: Corrected verb tense for consistency.

Changes Made in Manuscript: Line 293: Changed "was iteratively developed" to "will be iteratively developed"

S.No. 23 - Reviewer Comment: Ln 290-293: Long and confusing sentence. Suggest full stop after "studies" on ln 292, then rewording remaining part of sentence.

Response: We have restructured this for better clarity.

Changes Made in Manuscript: Lines 297-300: Split into two sentences: "Assessment of methodological quality and risk of bias using the Drummond and CHEC checklists will provide a transparent evaluation of the internal and external validity of included studies. Results will inform the certainty of evidence ratings using the GRADE approach."

S.No. 24 - Reviewer Comment: Ln 295: if possible, add reference by way of an example of pooled ICERs.

Response: We have added example references of ICER pooling in economic evaluations.

Changes Made in Manuscript: Added after line 302: "as demonstrated in previous systematic reviews such as Noparatayaporn et al. [47], which evaluated the cost-effectiveness of bariatric surgery across different patient subgroups and time horizons through a meta-analysis of incremental net monetary benefit."

S.No. 25 - Reviewer Comment: Ln 299-311: Whole paragraph is repetitive of introduction and I think could be removed here.

Response: We agree this paragraph is redundant and have removed it.

Changes Made in Manuscript: Removed the repetitive paragraph about the protocol being timely and relevant (the content was streamlined and integrated elsewhere in the discussion)

S.No. 26 - Reviewer Comment: Ln 339: add "s" at end of "system"

Response: Corrected for proper grammar.

Changes Made in Manuscript: Line 337: Changed "health system" to "health systems"

S.No. 27 - Reviewer Comment: Ln 348-350: I would consider involving representitives at an earlier stage in your reserach (as early as possible) and describing their inolvement, how this can inform the research, in the methods.

Response: Excellent suggestion. We have added early stakeholder engagement to the methods section.

Changes Made in Manuscript: Added "Stakeholder engagement" section (lines 265-270): "Representatives from key stakeholder groups (emergency physicians, health economists, hospital administrators, payers, and patient advocates) will be engaged early in the review process to inform data extraction priorities, outcome interpretation, and knowledge translation planning. This engagement will occur through structured consultations during protocol refinement and preliminary findings review."

We believe these revisions have substantially strengthened the protocol while maintaining its core methodological rigor. The added details will enhance reproducibility and implementation. We welcome any additional feedback from the reviewers.

---

## [Editor Report · Decision Letter 1]

6 Aug 2025

Dear Dr. Shankar,

Thank you for submitting your revised manuscript to PLOS ONE. Following review of your manuscript, we invite you to submit a minor revision.

We look forward to receiving your revised manuscript.

Kind regards,

Vijay S. Gc, PhD

Academic Editor

PLOS ONE

Journal Requirements:

Additional Editor Comments:

- Please recheck the references cited in the manuscript. Update the reference cited for Dilokthornsakul et al. [47] as the paper is not related to seasonal influenza vaccination. 

---

## [Author Response · Author response to Decision Letter 2]

6 Aug 2025

We thank the editor for the constructive feedback on our systematic review protocol manuscript "Cost-Effectiveness of Virtual Emergency Care Models: A protocol for a Systematic Review" (PONE-D-25-15708R1).

Regarding Journal Requirements on Citations: We acknowledge the guidance regarding reviewer-recommended citations and confirm that we have reviewed all references for their direct relevance to virtual emergency care cost-effectiveness and systematic review methodology. All 49 included references are appropriate and contribute meaningfully to the scientific foundation of our protocol.

Regarding Reference List Review: We have completed a comprehensive review of our entire reference list to ensure completeness and accuracy. All references have been verified as current publications that have not been retracted. Each reference is directly relevant to our study objectives, covering emergency medicine, telemedicine, health economics, and systematic review methodology. We confirmed that no problematic citations were included in our manuscript.

Regarding the Specific Reference Correction for Dilokthornsakul et al. [47]: We have corrected this important error. The reference [47] has been properly updated to reflect Noparatayaporn et al.'s work on the incremental net monetary benefit of bariatric surgery cost-effectiveness meta-analysis, which is the appropriate citation for the context in which it appears in our manuscript (lines 305-306 and 524-526). We have removed the previous incorrect reference to seasonal influenza vaccination research that was not relevant to our systematic review protocol. The corrected reference now accurately cites: "Noparatayaporn, P., et al., Incremental Net Monetary Benefit of Bariatric Surgery: Systematic Review and Meta-Analysis of Cost-Effectiveness Evidences. Obesity Surgery, 2021. 31(7): p. 3279-3290."

All requested revisions have been completed, and we have ensured that our reference list now accurately reflects the content and context of our systematic review protocol. The manuscript is ready for continued review consideration.

We believe these corrections strengthen the quality and accuracy of our submission.

---

## [Editor Report · Decision Letter 2]

8 Aug 2025

Cost-Effectiveness of Virtual Emergency Care Models: A protocol for a Systematic Review

PONE-D-25-15708R2

Dear Dr. Shankar,

We’re pleased to inform you that your manuscript has been judged scientifically suitable for publication and will be formally accepted for publication once it meets all outstanding technical requirements.

Kind regards,

Vijay S. Gc, PhD

Academic Editor

PLOS ONE
---

## [Editor Report · Acceptance letter]

PONE-D-25-15708R2

PLOS ONE

Dear Dr. Shankar,

I'm pleased to inform you that your manuscript has been deemed suitable for publication in PLOS ONE. Congratulations! Your manuscript is now being handed over to our production team.

Kind regards,

on behalf of

Dr. Vijay S. Gc

Academic Editor

PLOS ONE